# Recent Advances in the Synthesis of Aromatic Azo Compounds

**DOI:** 10.3390/molecules28186741

**Published:** 2023-09-21

**Authors:** Meng-Yun Zhao, Yue-Feng Tang, Guo-Zhi Han

**Affiliations:** College of Chemistry and Molecular Engineering, Nanjing Tech University, Nanjing 211816, China; zhaomengyun@njtech.edu.cn (M.-Y.Z.); 202161105021@njtech.edu.cn (Y.-F.T.)

**Keywords:** review, aromatic azo compounds, advances, synthesis, functional nanomaterials

## Abstract

Aromatic azo compounds have -N=N- double bonds as well as a larger π electron conjugation system, which endows aromatic azo compounds with wide applications in the fields of functional materials. The properties of aromatic azo compounds are closely related to the substituents on their aromatic rings. However, traditional synthesis methods, such as the coupling of diazo salts, have a significant limitation with respect to the structural design of aromatic azo compounds. Therefore, many scientists have devoted their efforts to developing new synthetic methods. Moreover, recent advances in the synthesis of aromatic azo compounds have led to improvements in the design and preparation of light-response materials at the molecular level. This review summarizes the important synthetic progress of aromatic azo compounds in recent years, with an emphasis on the pioneering contribution of functional nanomaterials to the field.

## 1. Introduction

Azo compounds usually refer to compounds containing -N=N- double bond, which can be divided into aryl azo compounds and alkyl azo compounds depending on the substituents on the -N=N- double bonds, as well as symmetrical and asymmetrical azo compounds from the point of view of structure. Azo compounds date back to 1859, and the high reactivity of -N=N- double bonds endow azo compounds with wide applications in many fields such as organic dyes, radical reaction initiators, and so on [1,2,3,4,5,6,7,8,9,10,11,12,13,14,15,16,17,18,19,20,21,22,23,24,25,26,27,28,29,30,31,32]. On the other hand, azo compounds have cis and trans isomers, which can convert to each other under light irradiation or heating, as was discovered as early as 1937. The special property further promotes the pioneering role of azo compounds in the field of optical functional materials. Among azo compounds, aromatic azo compounds have a higher π electron conjugation system along with higher chemical stability and thermal stability. Moreover, the substituents on the aromatic ring have a direct impact on their properties, which promoted the design of functional photoresponsive materials at the molecular level [33,34,35,36,37,38,39,40,41,42,43,44,45,46,47,48,49,50,51,52,53,54,55,56,57,58,59,60,61,62,63,64,65,66,67,68,69,70,71,72,73,74]. As a result, the research and application of aromatic azo compounds have received great attention and undergone significant development in the past ten years, as shown in Figure 1.

Generally, there are two traditional methods for synthesizing aromatic azo compounds, as shown in Figure 1: one is the coupling reaction of diazo compounds with electron-rich aromatics, which has a fast reaction rate and high yield—however, the substrate scope of the method is narrow, with a low safety factor; another method is the Mills reaction, which produces aromatic azo compounds from nitroso aromatic compounds and primary aromatic amines catalyzed by acetic acid. Therefore, the development of green and efficient methods for the synthesis of aromatic azo compounds is a significant concern in the chemical community. In this paper, we summarized the important synthetic progress of aromatic azo compounds in recent years, with an emphasis on the pioneering contribution of functional nanomaterials to the field.

## 2. Advances in the Synthesis of Aromatic Azo Compounds

### 2.1. Direct Oxidation of Aromatic Amines and Their Derivatives

The direct oxidation of aromatic amines to form aromatic azo compounds has been shown to be a green and promising method due to the wide availability of raw materials with rich and diverse structures, overcoming many shortcomings of traditional synthesis methods. For the oxidant in this strategy, oxygen is the most ideal choice for the green and atomic economy. However, oxygen molecules usually require activation via certain catalyst to participate in this reaction, which limits the direct utilization of oxygen in this process. In the past decade, transition metal compounds have entered people’s awareness due to their variable valence states and redox potentials. Furthermore, catalyst circulation can be achieved by the oxygen in the air. For example, Dutta et al. developed a facile, cost-effective method by which to synthesize diverse symmetrical and unsymmetrical aromatic azo compounds with inexpensive mesoporous manganese oxide materials as the catalyst and air as the terminal oxidant. Under the atmospheric condition, a variety of aniline derivatives underwent oxidative homo-coupling or cross-coupling to form corresponding azo compounds with moderate-to-excellent yields. Mild reaction conditions with good reusability endow the catalytic protocol with strong application prospects. Mechanism research indicated that air played a key role in the process, as shown in Figure 2 [75]. 

The extensive use of solvents in traditional organic chemicals is an important cause of environmental problems; therefore, developing solvent-free reactions has long been an attractive research direction. In our previous work, we report a new one-step direct synthesis of aromatic azo compounds from anilines under mild conditions. With the catalysis of copper acetate assisted by a small amount of palladium salt, rapid conversion of anilines to aromatic azo compounds can be observed under base-free and solvent-free conditions. In addition, the cross-coupling reaction based on this strategy also realized satisfactory yields. In this strategy, copper ions play a key role in the catalytic cycle via the auxiliary effect of palladium salt and oxygen, as shown in Figure 3 [76]. This method provides not only a new green route for the synthesis of symmetric and asymmetric aromatic azo compounds but also a strategy for exploring the catalytic applications of transition metal compounds.

Intramolecular diazotization reactions are often a challenge due to the issues of steric hindrance and angular tension. Maier et al. reported a new method for the synthesis of novel cyclic azo benzenes using diarylamine as the raw material, as shown in Figure 4. *m*-CPBA (Dichloromethane) was used as the oxidant and HOAc (Acetic acid)/DCM (Dichloromethane) as the mixed solvent [77]. The discovery of cyclic azobenzenes provides a novel photomolecular switch. Diarylamines with strong electron-withdrawing groups and electron-donating groups as feedstock reduce yields. Amino-substituted diarylamines require later derivatization in order to react. In addition, the electronic properties of substituent groups only had a weak effect on the yields of the product. The main disadvantages of this method are limited substrate selectivity and the possibility of generating multiple by-products during the reaction.

As mentioned above, for the strategy of the direct oxidation of aromatic amines to form aromatic azo compounds, the control of the oxidation degree is an important issue; that is, the catalytic oxidation of aniline usually produces by-products such as nitrobenzene, azobenzene, and azobenzene oxide, which are closely related to the performance of the catalyst. Shukla et al. developed a kind of Cu-CeO_2_ nanoparticle via a one-pot method; then, using H_2_O_2_ as the oxidant and acetonitrile as the solvent, direct oxidation of aniline to form aromatic azo compounds with high yields and selectivity was realized, as shown in Figure 5 [78]. Under the optimized conditions and with 3.8% Cu-CeO_2_ catalyst, 95% conversion of aniline with 92% selectivity of azo benzene was obtained. In particular, the amount of Cu and Ce present in the spent catalyst was found to be almost the same as that of the pristine catalyst, which confirmed the stability and heterogeneity of the catalyst. 

Alkyl 2-phenylazocarboxylates are a kind of asymmetric azo compound which can play a key role in various organic reactions such as the catalytic Mitsunobu reaction. However, the conditions for the synthesis of alkyl 2-phenylazocarboxylates are always very harsh. Kim et al. reported a route for the preparation of alkyl ethyl 2-phenylazocarboxylate from the oxidation of ethyl 2-Phenylhydrazinecarboxylate catalyzed by CuCl and DMAP (4-dimethylaminopyridine) in the presence of air [79], and yields of up to 95% were obtained within three hours (Figure 6). Furthermore, one advantage of this method is its strong tolerance to solvents and chlorinated solvents such as chloroform, dichloroethane, and dichloromethane, showing overall excellent reactivity. 

Trichloroisocyanic acid (TCCA) is an excellent oxidizing agent due to its stability, harmlessness, and ease of handling. Su et al. used trichloroisocyanuric acid as an oxidizing agent to oxidize phenylhydrazine compounds for the preparation of azo compounds [80]. Under the optimized conditions, the highest yield of 97% was obtained. The mechanistic diagram of this reaction is shown in Figure 7. In addition, the method easily realizes gram-scale synthesis with a less restrictive substrate range, except for that with large steric hindrance. 

Although there has been significant progress in the synthesis of aromatic azo compounds, there are relatively few reports on the construction of heteroaromatic azo derivatives. Jiang et al. reported a new method for the synthesis of azo compounds via pyrazol-5-amine iodination using *t*-butyl hydroperoxide (TBHP) as the oxidizer and copper salt as the catalyst [81]. The single-electron transfer (SET) mechanism (through oxidation of the reaction) is shown in Figure 8. Intermolecular iodination and oxidation simultaneously form C-I and N-N bonds, followed by oxidative dehydrogenation to synthesize azopyrroles and iodo-substituted azopyrroles with a wide diversity in substituents. The radical initiator of TBHP was essential for this transformation. Furthermore, the selective formation of highly functionalized heteroaromatic azo compounds can be controlled by the catalytic system. The mild reaction conditions, selective modification of pyrrole skeleton, and high bond-forming efficiency (BFE) endow this strategy with high value. The drawback of low yield in the derivatization reaction needs to be optimized for practical applications.

Overall, the direct oxidation of aromatic amines to synthesize azo compounds has many advantages; however, sometimes, the presence of peroxidation leads to more by-products. Additionally, some oxidants are not suitable for large-scale applications. 

### 2.2. Reductive Coupling of Aromatic Nitro Compound

An important source of aromatic amine is the reduction of aromatic nitro compounds. Therefore, starting from aromatic nitro compounds to synthesize aromatic azo compounds in one step will greatly improve the economy and environmental friendliness of reactions. In recent years, this synthesis strategy has attracted widespread attention from the scientific research community. Mondal et al. prepared a kind of AuNPs using a discrete a nanoscale organic cage (OC1^R^) as a template (Au@OC1^R^) [82]. The cage-immobilized AuNPs can act as heterogeneous photocatalyst for the selective reduction of nitroaromatics by 2-propanol to form the corresponding azo compounds with high yields at room temperature, as shown in Figure 9. After optimizing the synthesis conditions, the corresponding azo compounds could be selectively obtained, with 99% conversion under UV irradiation for 2 h in an inert atmosphere. In addition, no azo compounds were produced in the absence of AuNPs or only in the presence of OC1^R^, which indicated that the AuNPs is crucial for the reaction. Furthermore, the OC1^R^ endows the catalyst with the advantages of easy separation and good compatibility of functional groups. The work lays an innovative foundation for the development of a new strategy for the synthesis of azo compounds. 

Due to quantum size effects, Pd nanoclusters (PdNCs) with diameters less than 2 nm exhibit better catalytic properties than ordinary Pd nanoparticles. Generally, these ultra-small PdNCs must be complexed with specific ligands to maintain stable morphology. Yan et al. reported a tandem reduction strategy for the selective conversion of nitroaromatics to five types of products—aniline, hydroxylamine, azoxybenzenes, azo compounds, and hydrazine compounds—under mild conditions, as shown in Figure 10 [83]. First, Pd(OAc)_2_ was in situ reduced by NaBH_4_ to form ultra-fine PdNCs. These ultra-fine PdNCs were stabilized via surface-ligating with nitroaromatics and uniformly dispersed in the solvent. Then, the selective reduction of nitroarene was catalyzed by the ultra-fine PdNCs. Products with the electron-donating group and the electron-withdrawing group were also obtained with high yields. In addition, nitro fused aromatic compounds such as nitronaphthalene also adopt this protocol with strong results, i.e., a high yield of 80%, which was rarely reported before. However, the hydroxylamine generated in the reaction is easily oxidized, which may produce side reactions, and further optimization of the reaction conditions is needed to improve the selectivity of the reaction.

At present, new functional materials such as boron nitride (BN) are flourishing in the field of catalysis. Similar to graphene, hexagonal boron nitride (h-BN) is a two-dimensional layered material with low toxicity and thermal stability. For catalytic application, h-BN can provide a supported platform and more reactive sites for metal nanoparticles. Liu et al. synthesized a kind of Au nanoparticle loaded on h-BN nanoplates (Au/BN). The composite catalyst can selectively catalyze the conversion of nitrobenzene to azobenzene or hydrogenated azobenzene in the presence of IPA (*i*-propyl alcohol)/KOH under N_2_ or air atmosphere, as shown in Figure 11 [84]. In the process, h-BN inhibited the activation of oxygen, allowing the catalytic hydrogenation of nitrobenzene in air. On the other hand, KOH takes away the hydrogen atom from the isopropanol, subsequently producing acetone and an activated H donor on the Au/BN surface. Au nanoparticles are then bound to the H donor to form H-Au. The active species of H-Au played a key role in the subsequent reduction process. Meanwhile, the H atoms in H-Au can collide with each other to produce H_2_ during the reaction for the next cycle. 

Although the catalytic activity of noble metal nanoparticles is often excellent, the high cost has hindered their large-scale use. Therefore, other relatively inexpensive transition metals catalysts have attracted great attention. Pahalagedara et al. reported a sea urchin-like Ni/graphene nanocomposite for the selective reduction by hydrazine of nitroaromatics with different substituents to the corresponding azo compounds, and the magnetic catalyst was easily recycled and reused. In addition to stabilizing and dispersing nanoparticles, graphene can improve the contact between the reactants and the catalyst surface by interacting with nitroaromatics through p-p stacking, thus increasing the reaction rate, as shown in Figure 12 [85]. In the reduction process, hydrazine was first oxidized to produce the electrons necessary for the reduction, along with nitrogen and water. Then, the nitrobenzene was reduced to nitrosobenzene, which was further reduced to *N*-phenylhydroxylamine. The nitrosobenzene and *N*-phenylhydroxylamine were finally condensed to form the main product of azoxybenzene and azoxybenzene. 

Wang et al. prepared a kind of Fe and N co-doped mesoporous carbon (NMC-Fe) as an efficient heterogeneous catalyst for the reduction of nitroaromatics to azo compounds via hydrazine hydrate, as shown in Figure 13 [86]. Doped N and Fe occupied vacant and defective sites in carbon nanosheets, which resulted in a smaller specific surface area than undoped carbon catalysts. Though both Fe-doped (MC-Fe) and N-doped (NMC) carbon materials can catalyze the reduction of 1,2-bis(4-chlorophenyl) diazene oxide to form (E)-1,2-bis(4-chlorophenyl) diazene, the Fe and N co-doped strategy endowed the reaction with higher selectivity. Typically, iron-based catalyst involve a hydrogen transfer mechanism. Negatively charged hydride was adsorbed at the iron active center (the electron-withdrawing group). The formed complexes act as hydrogen transfer centers for the selective reduction of nitroaromatics to azo compounds. However, in the absence of Fe or N, only azoxybenzene was obtained. 

Compared to other transition metals, Cu is cheaper and more available and easier to handle. Moran et al. reported efficient copper nanoparticles (Cu(0)NPs) for the catalysis of nitroaromatics with the controlled and selective transfer of hydrogenation to prepare azo compounds through different hydrogen sources, as shown in Figure 14 [87]. The highlight of this work was that different hydrogen donors gave different products. For example, using ethanolamine as the hydrogen source, azo benzene was obtained with 96% selectivity after 25 h at 55 °C, and study of the substrate’s scope provided azo derivatives, with an average yield of 85%. However, using glycerol as the source of hydrogen, the end product was almost always aniline. Moreover, the nano-Cu material suppressed the rate of auto-oxidation, and only a trace of Cu_2_O was detected after 6 months, as well as the still-high catalytic activity. 

As a hydrogen source for reduction reactions, hydrogen is undoubtedly a green and low-cost choice for industrialization. Hu et al. reported the preparation of the azo compounds from nitroaromatics under mild conditions catalyzed by a worm-like Pd nanomaterial. The diameter of the Pd catalyst was about 3.5 nm with a narrow size distribution. The worm-like Pd nanomaterial can catalyze the reduction of nitrobenzene to form azo-benzene via H_2_ in the presence of a base, whereas it forms aniline in the absence of a base. It is worth mentioning that higher yields were obtained for the electron-rich nitroaromatic compounds. In addition, asymmetric azo compounds were facilely synthesized by this method, with good yields [88]. The plausible mechanism proposed by the authors is shown in Figure 15. First, hydrogen was adsorbed on the surface of the palladium nanoparticles and reduced nitrobenzene to form nitrosobenzene, which, in turn, rapidly converted to *N*-phenylhydroxylamine. Under acidic or neutral conditions, *N*-phenylhydroxylamine was further reduced to aniline. Under alkaline conditions, N-phenylhydroxylamine combined with nitrosobenzene to form *N*, *N*′-dihydroxy-diphenylhydrazine, which was then further reduced to azobenzene. Moreover, excessive reduction products of hydrazobenzene can be spontaneously oxidized to azobenzene in air.

Huang et al. prepared a kind of cobalt/nitrogen-doped carbon (IPA_x)_ catalyst for the selective reduction of nitroaromatics to aniline or aromatic azo compounds by H_2_, which depends on the basicity of the reaction system [89]. Furthermore, recycling experiments showed that the Co-N_x_ catalyst had excellent reusability. The mechanism of this reaction is shown in Figure 16. Similar to the classical reduction of nitrobenzene, nitrobenzene was first hydrogenated to form nitrosobenzene, followed by transfer to *N*-phenylhydroxylamine and finally aniline. However, the hydrogenation pathway of nitrobenzene was changed under basic conditions. The activating energy of the condensation reaction between *N*-phenylhydroxylamine and nitrosobenzene was reduced, as was the pathway in which *N*-phenylhydroxylamine transfer to aniline was inhibited. Therefore, azoxybenzene was readily produced.

Compared to the direct oxidative coupling of aromatic amines, the synthesis of aromatic azo compounds, through the reduction coupling of aromatic nitro compounds, undoubtedly has higher efficiency. However, the catalysts involved often have high costs, which limits their practical application prospects.

### 2.3. Electrochemical Method

Organic electrochemical synthesis has become a practical and environmentally friendly synthesis method and is widely used in oxidation and reduction reactions. In electrochemical synthesis, electrodes act as acceptors or donors of electrons to avoid, to a certain extent, the use of some harmful and dangerous chemical oxidation and reduction reagents, as well as to reduce the generation of chemical waste and improve production safety. Therefore, the electrochemical synthesis of some fine organic chemicals has attracted great attention in recent years. Qiao et al. first prepared a Ni_3_Fe-MOF-OH material with surface hydroxylation. Then, using the material as electrodes, azobenzene was synthesized via cathodic reduction of nitrobenzene and anodic oxidation of aniline, as shown in Figure 17 [90]. Experimental results indicated that the bimetallic Ni_3_Fe-MOF-OH electrocatalyst showed excellent performance in N-N coupling. In the process, the surface hydroxylation of the electrodes promoted the adsorption of nitrobenzene and aniline and improved the reaction rate. On the other hand, the competitive hydrogen and oxygen evolution reactions were suppressed due to the adsorption of nitroarenes and anilines via surface hydroxyls of the electrocatalyst. Moreover, using TEMPO as the electron medium, gram-scale reactions were realized with high selectivity. 

Gong et al. prepared a kind of N-doped carbon nanotube-supported Ni-Co alloy nanoparticle (NiCo@N-CNTs) via a simple reductive pyrolysis strategy for the electrochemical synthesis of azobenzene [91]. In the Ni–Co-alloy nanoparticles, CoNPs confined at the tip of N-doped CNTs (N-CNTs) showed excellent activity and thermal stability toward thermochemical selective hydrogenation of aldehyde, ketone, carboxyl, and nitro groups. It was found that when Ni-Co@N-CNTs acted as cathodes, 100% conversion with 99% selectivity of oxidized azobenzene were achieved. Furthermore, in order to improve the energy utilization efficiency, the authors designed a NiCo@N-CNTs||Ni(OH)_2_/NF dual electrode electrolyzer, as shown in Figure 18. Simultaneous cathodic reduction of nitrobenzene and anodic oxidation of 5-hydroxymethylfurfural were achieved, with high yields. This work provides a new idea for the design and fabrication of highly active, durable, and low-cost electrocatalysts for other electrocatalytic syntheses.

Zhang et al. reported a method for the cross-coupling reaction of aromatic nitro compounds to aromatic azo compounds via base-free electrochemistry using SmI_2_ as a catalyst, as shown in Figure 19 [92]. Under the optimized conditions, desired asymmetric azo compounds were synthesized, with 83% yield and 99% selectivity. In this strategy, the electron-donating groups exhibited better adaptability. Moreover, the samarium electrode was rarely consumed in the reaction process and can be reused more than 100 times. The preliminary mechanistic study suggested that the formation of azobenzene was accomplished by successive single-electron reductions. The key step was the reduction of nitro benzene to a radical anionic intermediate. The rapid dimerization of this intermediate to produce oxo azobenzene was followed by a single-electron transfer reduction to form the desired product mediated by Sm^II^X_2_.

Although electrochemistry has many advantages, there are still some issues that need to be addressed. For example, in the process of electrochemical reaction, the electron transfer between the electrode and the substrate is heterogeneous, which would lead to overpotential and make the functional group tolerance of the reaction worse. 

### 2.4. Photocatalytic Method

Photocatalytic synthesis is an emerging method in the organic field, in which the catalyst utilizes light energy to promote organic reaction. Compared with traditional organic synthesis, it has the advantages of mild reaction conditions, fast reaction speed, high selectivity, and lower energy consumption. Zhou et al. reported that Pd nanoparticles loaded on mesoporous CdS (Pd@ CdS) can act as a highly active and selective photocatalyst in water under visible light irradiation [93]. The photocatalytic conversion of glucose to arabinose and nitrosobenzene to azobenzene could be carried out simultaneously, catalyzed by the composite catalyst with ideal selectivity, as shown in Figure 20. In this process, photoexcited electrons were transferred from mesoporous CdS to PdNPs, thus inhibiting the recombination of electron–hole pairs and providing the active site on Pd for the reduction of nitrosobenzene to azobenzene, while glucose was photo-oxidized by holes to arabinose through C1-C2 bond cleavage. Moreover, the formic acid generated by glucose oxidation was conducive to the hydrogenation reaction of nitrobenzene. On the other hand, nitrobenzene promotes the conversion of glucose by accepting photoexcited electrons and H^+^ generated from water cleavage and glucose reforming.

Wang et al. reported a kind of CQDs/ZnIn_2_S_4_ nanocomposite for the controlled hydrogenation of nitrobenzene under visible light. By simply adjusting the alkalinity and hydrogen source, azobenzene or azoxybenzene were selectively produced [94]. When triethanolamine was used as the hydrogen source, 76% aniline was obtained. However, after a large amount of base was added, azobenzene was mainly produced, as shown in Figure 21. When irradiated by visible light, ZnIn_2_S_4_ was excited to generate electrons and holes, and CQDs (carbon quantum dots) transfer photogenerated electrons away from ZnIn_2_S_4_. With the assistance of triethanolamine (TEOA) as the hydrogen source, the electron-rich CQDs promote the gradual hydrogenation of nitrobenzene to aniline. Meanwhile, TEOA also acts as an electron donor, which reacts with photogenerated holes in ZnIn_2_S_4_ to complete the whole photocatalytic cycle. The introduction of bases such as NaOH into the reaction system promotes the condensation of nitrosobenzene with *N*-phenylhydroxylamine to form azoxybenzene, which can be further hydrogenated to produce azobenzene. Despite many advantages, photocatalytic synthesis still faces challenges such as low quantum efficiency, low solar energy utilization, and insufficient catalyst recycling rate. 

Overall, photocatalysis is one of the key technologies currently being developed, and the design and synthesis of core catalysts are the most important factors. Catalysts based on noble metals are clearly not suitable for the promotion and application of this technology.

### 2.5. Biochemical Methods 

Entering the 21st century, humanity has been facing an unprecedented crisis of survival and development due to continuous depletion of fossil fuels and increasing environmental pollution. Therefore, the traditional chemical industry must undergo revolutionary transformation. Biochemistry is just one of the technologies needed for the sustainable development of society. In the field of organic synthesis, the core of biochemistry is to use enzymes to replace traditional industrial catalysts, to promote chemical reactions with high efficiency and selectivity, and simultaneously, to reduce energy consumption and environmental pollution. Sousa et al. reported the synthesis of azo compounds via the oxidative coupling of primary aromatic amines catalyzed by laccase under mild reaction conditions, as shown in Figure 22 [95]. The enzyme first promoted the aerobic oxidation of amine to form an unstable intermediate of amino cation radical, which subsequently underwent deprotonation to produce an amino neutral radical. Then, two free radicals were coupled to form HN-NH bonds, which were dehydrogenated to give the corresponding azo compounds. However, there is one drawback of this strategy that cannot be ignored: for substrates containing electron-withdrawing groups, unstable radical intermediates may be generated, which results in low yields.

Ethyl lactate is a biodegradable and environmentally friendly biomass-derived solvent which is completely degradable, having non-toxicity and low-corrosivity. Pariyar et al. used ethyl lactate as a solvent for the facile one-step synthesis of symmetrical and asymmetrical aromatic azo compounds from amines [96]. In the strategy, ethyl lactate was used as an effective mediator to synthesize azobenzene from aniline under catalyst-free conditions, as shown in Figure 23. The oxidation of aniline by potassium peroxomonosulfate (oxone) is possibly conducted via a free radical mechanism or electrophilic oxygen transfer, producing the unstable nitrosobenzene. Subsequently, another molecule of aniline attacks the nitrosobenzene, followed by dehydration to give trans-azobenzene. In the process, ethyl lactate acted as a biological enzyme. Furthermore, the amines with electron-donating groups gave higher yields than those with electron-withdrawing groups. The most important advantage of this method is the use of environmentally friendly solvents, which act as mimic enzymes. However, the mechanism has not been elucidated yet, and this method is currently not suitable for the synthesis of asymmetric azo compounds. 

### 2.6. Nitrogen–Halogen Exchange 

In recent years, construction of -N=N- bonds using hydrazine hydrate as an inorganic dinitrogen source has received much attention, being a direct and green pathway for the synthesis of aromatic azo compounds. Xie et al. reported a copper-catalyzed diarylation reaction of hydrazine with cyclic/linear diaryl iodonium salts, which converted inorganic hydrazine into organic nitrogen-containing compounds to obtain a series of azobenzene derivatives, as shown in Figure 24 [97]. In this strategy, phthalhydrazide (PHA) was first formed via the hydrazinolysis of phthalic anhydride. At the same time, the coordination of CuI with 2,2′-bipyridine yielded a kind of CuI-complex (Int-A). The next process has two possible pathways: (1) Int-A underwent ligand exchange with PHA, followed by an oxidative addition of cyclic diphenyliodonium triflate (2a) to produce Int-B and Int-D in sequence (pathway a); (2) Int-A undergoes direct oxidative addition with 2a to produce Int-C, followed by ligand exchange to form Int-D (pathway b). Next, reductive elimination of Int-A gives the monoaryl compound IM 1. A second oxidative addition of IM 1 with Int-A gives Int-E, prior to subsequent intramolecular ligand coupling to form Int-F. A second reductive elimination of Int-A releases compound IM 2, which then undergoes K_2_CO_3_-mediated deprotection and oxidation to give the azo compound and potassium phthalic acid.

*N*-aryl-*N*′-silyldiazenes are a kind of kinetically stable aryl nucleophilic reagent. Finck et al. synthesized a series of asymmetrical azobenzene derivatives via palladium-catalyzed C-N coupling reactions between *N*-aryl-*N*′-silyl diazenes and aryl halides, as shown in Figure 25 [98]. First, L_2_Pd^0^(I) underwent oxidative addition with X-Ar^2^ to produce aryl palladium(II) halide (II). Then, the intermediate II underwent σ-bond complexation with the *N*-aryl-*N*′-silyldiazenes to produce transition state III. Subsequently, transition state III released Me_3_Si-X to produce intermediate IV, which was finally reduced to give asymmetric aromatic aromatics as well as L_2_Pd^0^(I), thus completing the catalytic cycle. It is worth noting that the presence of the base is crucial, and Cs_2_CO_3_ was the optimized choice. 

Overall, the strategy of nitrogen–halogen exchange has shown good application prospects in the construction of asymmetric aromatic azo compounds, but the tolerance of functional groups and product yields still need to be further improved.

## 3. Conclusions

Azo aromatic compounds are important functional molecules and chemical raw materials which are widely used in the fields of organic dyes, optical materials, and so on. Due to the shortcomings of traditional synthesis methods, the design of new materials involving azo molecules at the molecular level requires significant improvements in synthesis methods. This paper reviews the recent research advances in the synthesis of aromatic azo compounds, including symmetric, asymmetric, and cyclic azo compounds, and with an emphasis on the pioneering contribution of functional nanomaterials to the field. We hope that this review will point out the advantages and limitations in the current applications of functional nanomaterials for the synthesis of aromatic azo compounds, and provide new ideas for the preparation of azo compounds and their derivatives. Although the catalysis of nanomaterials can overcome some of the shortcomings of traditional methods, their preparation and cost may limit their large-scale application. Therefore, finding cheaper nanomaterials and improving their recyclability will continue to be an important research direction.

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
