# Peer review of "Recent Advances in the Synthesis of Aromatic Azo Compounds"

_molecules, 2023, doi:10.3390/molecules28186741_

Round 1
Reviewer 1 Report
This review summarizes the important synthetic progress of aromatic azo compounds in recent years, with an emphasis on the pioneering contribution of functional nanomaterials in the field. Overall, the manuscript is able to summarize the relevant literature of recent years, but there are more problems could be considered. Specifically, the authors should address the following issues. 1. Most of the literature is simply reviewed in the manuscript and lacks effective evaluation, which can be considered in terms of concentration of Azo compounds; 2. As to the Fig. 1. I suggest the author have to analyze the Azo compounds in detail from its classify. 3. The manuscript claims to have explored the mechanism in the abstract, but in fact only individual applications of the literature have explored the mechanism
4. Comprehensiveness and completeness of the content: The content presented in the manuscript is generally comprehensive and complete. However, to improve the comprehensiveness and completeness of the content, you can use appropriate charts and tables and express some information in a more concise form
5. Make suggestions: Finally, you can make your own suggestions to improve the manuscript. For example, you can use more charts and tables to make the content more graphical and the manuscript more interesting. Also, you can express some information in a more concise form and put more emphasis on the key points of the manuscript. Finally, you can use newer and more reliable sources to make your content more up-to-date and accurate. Overall, this article is worth reading and studying, and I emphasize that by correcting some things, you can make improvements in its quality. Regarding the references, it is a list of acceptable and valid sources that have been used in the article. However, some resources may be outdated and need to be updated. I suggest you use newer and more reliable sources if needed. Overall, you could improve the quality and appeal of this article by correcting some items and updating the references.
6. Some related refs on Azo compounds may be updated, such as Molecules 2023, 28, 4507;
minor revision
Author Response
This review summarizes the important synthetic progress of aromatic azo compounds in recent years, with an emphasis on the pioneering contribution of functional nanomaterials in the field. Overall, the manuscript is able to summarize the relevant literature of recent years, but there are more problems could be considered. Specifically, the authors should address the following issues.
- Most of the literature is simply reviewed in the manuscript and lacks effective evaluation, which can be considered in terms of concentration of Azo compounds;
Response: Thank you very much for your comment, we have added some summaries to each section.
- As to the Fig. 1. I suggest the author have to analyze the Azo compounds in detail from its classify.
Response: Thank you very much for your comment. Figure 1 only emphasizes the importance of azo compounds. The main theme of the manuscript is to review the synthesis of aromatic azo compounds. Therefore, based on your comment, we have revised the title to make the core of the manuscript clearer.
- The manuscript claims to have explored the mechanism in the abstract, but in fact only individual applications of the literature have explored the mechanism.
Response: Thank you very much for your comment. We have added some mechanism discussion. However, synthetic mechanism is not mentioned in some literatures.
- Comprehensiveness and completeness of the content: The content presented in the manuscript is generally comprehensive and complete. However, to improve the comprehensiveness and completeness of the content, you can use appropriate charts and tables and express some information in a more concise form.
Response: Thank you very much for your comments, we have added some charts according to your suggestion.
- Make suggestions: Finally, you can make your own suggestions to improve the manuscript. For example, you can use more charts and tables to make the content more graphical and the manuscript more interesting. Also, you can express some information in a more concise form and put more emphasis on the key points of the manuscript. Finally, you can use newer and more reliable sources to make your content more up-to-date and accurate. Overall, this article is worth reading and studying, and I emphasize that by correcting some things, you can make improvements in its quality. Regarding the references, it is a list of acceptable and valid sources that have been used in the article. However, some resources may be outdated and need to be updated. I suggest you use newer and more reliable sources if needed. Overall, you could improve the quality and appeal of this article by correcting some items and updating the references.
Response: Thank you very much for your comments. According to your suggestions, we have made comprehensive revisions to the manuscript and updated the references.
- Some related refs on Azo compounds may be updated, such as Molecules 2023, 28, 4507;
Response: Thank you very much for your comment, we have updated the reference section.
Reviewer 2 Report
1. Mention the source of information for the Figure 1.
2. Cite the reference in the text line 47-62.
3. List of abbreviation must be added in the manuscript.
4. Line 98; pristine catalyst? Reference?
5. Enriched the conclusion section with future prospect.
6. How this manuscript is different from various other manuscript published in this area? Explain it.
7. The authors, in my opinion, should include a discussion section as well as a comparison and comprehensive explanation of the advantages and disadvantages of each method of synthesis.
8. Add some more methods in the manuscript as mentioned in (a). Synthesis 2020; 52(07): 1103-1112 DOI: 10.1055/s-0039-1690052; (b). https://doi.org/10.1021/acs.joc.7b03119.
Author Response
- Mention the source of information for the Figure 1.
Response: Thank you very much for your comment, we have labeled the data sources in Figure 1.
- Cite the reference in the text line 47-62.
Response: Thank you very much for your comment, we have added the corresponding references.
- List of abbreviation must be added in the manuscript.
Response: Thank you very much for your comment. We have added list of abbreviation at the end of the manuscript.
- Line 98; pristine catalyst? Reference?
Response: The pristine catalyst is a kind of Cu-CeO2 nanoparticles. Moreover,the amount of Cu and Ce present in the spent catalyst was found to be almost the same as that of the pristine catalyst, which confirmed the stability and heterogeneity of the catalyst.
- Enriched the conclusion section with future prospect.
Response: Thank you very much for your comment. we have enriched the conclusion section with future prospect.
- How this manuscript is different from various other manuscript published in this area? Explain it.
Response: One characteristic of this manuscript is that the pioneering contribution of functional nanomaterials is reviewed.
- The authors, in my opinion, should include a discussion section as well as a comparison and comprehensive explanation of the advantages and disadvantages of each method of synthesis.
Response: Thank you very much for your comment, we have added discussion to each method of synthesis.
- Add some more methods in the manuscript as mentioned in (a). Synthesis 2020; 52(07): 1103-1112 DOI: 10.1055/s-0039-1690052; (b).https://doi.org/10.1021/acs.joc.7b03119.
Response: Thank you very much for your comment, we have added these methods in the revised manuscript.
Reviewer 3 Report
In this review, the authors consider two main parts for preparation azo-compounds. The first is the synthetic part: an overview of the main successes in recent years. The second part focuses on the use of various functional derivatives of nanomaterials (nanoparticles, nanoclusters, nanoplates, nanocomposites). These two parts are an interesting highlight in this review. It is worth noting that this review is written in a clear language, everything is accessible and consistently explained. Therefore, I recommend the review for publication in the journal Molecules. However, there are some comments that the authors should correct to make the review look even better.
1) Pages 3,5,6,9 - The figures in the work perfectly complement and enhance this review. However, their chaotic location, unfortunately, spoils the whole impression. Fix this moment.
2) Since the name review is quite general, I would like the authors to work out their concept towards nanomaterials in the title.
3) The authors report methods for synthesis aryl-substituted azo compounds. There are also examples in heterocyclic variations, which the authors forgot to mention. Add relevant examples to this overview.
4) "We hope that this review will point out the advantages and limitations in the current applications of functional nanomaterials for the synthesis of aromatic azo compounds, as well as provide new ideas for the preparation of azo compounds and their derivatives.”. Describe in detail what advantages we are talking about.
5) Since the authors give concrete examples of obtaining a compounds, it is better to write the yields and conditions under each schema where possible (for synthesis some basic compounds).
Author Response
In this review, the authors consider two main parts for preparation azo-compounds. The first is the synthetic part: an overview of the main successes in recent years. The second part focuses on the use of various functional derivatives of nanomaterials (nanoparticles, nanoclusters, nanoplates, nanocomposites). These two parts are an interesting highlight in this review. It is worth noting that this review is written in a clear language, everything is accessible and consistently explained. Therefore, I recommend the review for publication in the journal Molecules. However, there are some comments that the authors should correct to make the review look even better.
1. Pages 3,5,6,9 - The figures in the work perfectly complement and enhance this review. However, their chaotic location, unfortunately, spoils the whole impression. Fix this moment.
Response: Thank you very much for your comments, we have updated these figures..
2. Since the name review is quite general, I would like the authors to work out their concept towards nanomaterials in the title.
Response: Thank you very much for your comment. Although this review emphasizes the application of nanomaterials, not all methods involve nanomaterials. So, the article title did not highlight the concept.
3. The authors report methods for synthesis aryl-substituted azo compounds. There are also examples in heterocyclic variations, which the authors forgot to mention. Add relevant examples to this overview.
Response: Thank you very much for your comment, we have added examples of heterocyclic azo compounds.
4. "We hope that this review will point out the advantages and limitations in the current applications of functional nanomaterials for the synthesis of aromatic azo compounds, as well as provide new ideas for the preparation of azo compounds and their derivatives.”. Describe in detail what advantages we are talking about.
Response: Thank you very much for your comment, we have added some discussion in the section.
5. Since the authors give concrete examples of obtaining a compounds, it is better to write the yields and conditions under each schema where possible (for synthesis some basic compounds).
Response: Thank you very much for your comment, we have added the optimized condition information for each scheme.
Round 2
Reviewer 2 Report
1. The manuscript is ok now.